# Longitudinal study of risk factors predicting cannabis use disorder in UK young adults and adolescents
Martine Skumlien [1,2] ✉, Darcy Jones[1], Claire Mokrysz[3], Rachel Lees[1], Kat Petrilli[2], Shelan Ofori[3], Will Lawn [4], H. Valerie Curran[3] & Tom P. Freeman [1]

## Abstract

**Background** Cannabis use disorder (CUD) affects up to 1 in 5 people who try cannabis. In order to develop effective interventions to prevent and treat CUD, it is important to identify the factors that predict it. This longitudinal study investigated whether eight potential risk factors predicted CUD levels at 12-month follow-up.

**Methods** Participants were 232 young adults (26-29 years) and adolescents (16-17 years) (48%/52% males/females) who took part in both the baseline and 12-month follow-up sessions of the London-based 'CannTeen' study. Half of the participants ($n$ = 117) used cannabis 1-7 days/week at baseline and the other half had used cannabis maximum 10 times in their life. CUD was measured with the Mini Neuropsychiatric interview for DSM-5 CUD, which was used to categorise participants into no, mild, moderate, or severe CUD levels. Ordinal logistic regression was used to explore whether baseline CUD (yes/no), age-group, gender, days/week of cannabis use, problematic alcohol use, problematic tobacco use, past-year negative life events, and the COVID-19 lockdown predicted 12-month CUD levels in the full sample and in only those who used cannabis minimum once per week at baseline.

**Results** Here we show that adolescent age (odds ratio = 3.26, $p < 0.001$) and CUD at baseline (odds ratio = 45.15, $p < 0.001$) predict higher CUD levels at follow-up. We do not find evidence to support associations with other factors. The same pattern of results is found when including only participants who used cannabis at baseline.

**Conclusions** CUD prevention and treatment should be targeted towards adolescents, who have a significantly greater risk of CUD than young adults.

## Plain language summary

Up to 1 in 5 people who try cannabis go on to develop cannabis use disorder (CUD), but we do not know why some people develop CUD while others do not. Here, we explored associations between eight potential risk factors measured at baseline and CUD levels 12 months later. These factors were baseline CUD, frequency of cannabis use, adolescent age (vs. young adult age), gender, problematic alcohol use, problematic tobacco use, negative life events, and COVID-19 lockdown. Adolescents and those with CUD at baseline were more likely to have higher CUD levels after 12 months. We did not find evidence that the other factors predicted CUD. Our results suggest that CUD prevention and treatment interventions should be targeted towards adolescents.

Cannabis has been used by humans for at least five millennia. Today, roughly 228 million people worldwide use cannabis for both recreational and medicinal purposes[1]. Like many other drugs, cannabis carries a risk of addiction, clinically defined as cannabis use disorder (CUD). The fifth edition of the Diagnostic and Statistical Manual of Mental Disorders (DSM-5) defines CUD as problematic cannabis use leading to clinically significant distress and/or impairment in functioning. Symptoms include using more cannabis than intended, an inability to stop use despite harm, and tolerance and withdrawal symptoms. Individuals with CUD may also experience reduced quality of life alongside a higher risk of mental health, cognitive, and

physical problems, such as mood and psychotic disorders[2]. Although the estimated prevalence varies between different studies, the best available evidence suggests that between 9%[3] and 22%[4] of those who try cannabis eventually go on to develop CUD. Therefore, CUD affects a substantial minority of people who use cannabis (PWUC), and it is important to understand which risk factors predict CUD development.

The number of people seeking treatment for CUD in Europe has increased over the past 15 years, placing a major strain on healthcare services[5,6]. While behavioural and psychosocial interventions show some efficacy, at least in the short term, there are no approved pharmacotherapies

[1]Addiction and Mental Health Group (AIM), Department of Psychology, University of Bath, Bath, UK. [2]Department of Addictions, King's College London, London, UK. [3]Clinical Psychopharmacology Unit, University College London, London, UK. [4]Department of Psychology, King's College London, London, UK. ✉e-mail: ms4068@bath.ac.uk

for CUD[7]. In addition to this burden within treatment services, the vast majority of PWUC meeting criteria for CUD do not enter treatment[8,9]. Improving the efficacy and maximising the cost/benefit ratio of treatment and prevention measures requires the ability to identify those most at risk, so that interventions targeting these groups can be developed. In addition to using cannabis frequently and at large quantities, male gender (odds ratio, OR~2), adolescent age (OR~3), and more problematic use of other substances such as alcohol (OR~1.5-2.5) and tobacco (OR~4, based on one study) have all been identified as risk factors for CUD[2,10–13]. However, findings have not been consistent across studies, possibly due to differences in design, samples, and measurement tools, and up-to-date replication in different populations and settings is necessary.

Other potential risk factors are even less well established. For instance, while adverse childhood experiences, such as parental separation, are a relatively consistent predictor of problematic cannabis use and CUD[14–16], it is less clear if negative experiences later in life, such as losing one's job, increase CUD risk. One major global event, which was experienced as highly negative and stressful by many people, was the coronavirus disease (COVID) pandemic. The lockdown and social distancing measures implemented in many countries to mitigate the spread of the virus led to increased isolation, which is an important risk factor for mental health problems[17]. Several systematic reviews have suggested that the prevalence of anxiety, depression, and general psychological distress increased during the pandemic[18–21], especially in the early months of lockdown[22]. Longitudinal studies that started before the COVID-19 pandemic and continued during lockdown have an unprecedented opportunity to explore the effects of this global public health crisis.

The current longitudinal study investigates whether eight potential risk factors predict CUD levels after one year in both adolescents and young adults who do and do not use cannabis regularly and using a rigorous definition of CUD according to current DSM-5 diagnostic criteria. Our pre-registered[23] hypotheses were that male gender, younger age, greater cannabis use frequency, baseline CUD, problematic alcohol use, problematic tobacco use, more negative life events, and COVID-19 lockdown would be positively associated with greater CUD levels at the one-year follow-up. We replicate previous findings showing that adolescent age, as well as baseline CUD, are associated with follow-up CUD. We do not find any longitudinal relationship between CUD and the other relatively well-established risk factors gender, cannabis use frequency, and problematic alcohol and tobacco use, or the less explored risk factors negative life events and the COVID-19 lockdown.

## Methods
### Cannabis research context[24]
This study was conducted in London, United Kingdom, between 2017 and 2020. Recreational cannabis is illegal in the UK under the Misuse of Drugs Act 1971. Medical cannabis was legalised in 2018 and is regulated by the Misuse of Drugs Regulations 2001. The most common method of consumption in the UK during the study period was smoking cannabis in a joint together with tobacco and the most prevalent type of cannabis was sinsemilla or 'skunk', which contains 14-15% $\Delta^9$-tetrahydrocannabinol and negligible levels of cannabidiol[25–29]. The past-year prevalence of cannabis use in adults was estimated at 7.8% in 2019/2020, the highest since 2007 (8.2%)[30]. In 2019/2020, 19.6% of first-time presentations to substance use treatment among adults was for cannabis[31].

### Design
This is a longitudinal observational study using data from the baseline and 12-month follow-up sessions of the CannTeen study[32].

### Participants
The CannTeen study included 274 participants recruited from the Greater London area via school assemblies, physical posters and flyers, and social media advertisements. Of these, 234 (85.4%) completed the follow-up assessment, and 232 had complete data and were included in the study. The

two excluded participants had missing values on the negative life events measure (explained below).

The overall aim of the CannTeen study was to explore whether associations between cannabis use and multiple psychological, cognitive, and biological factors differed between adolescents and young adults. Therefore, participants were purposively recruited into four groups based on their age and cannabis use at baseline: adolescent PWUC ($n = 61$), young adult PWUC ($n = 56$), adolescent controls ($n = 57$), and young adult controls (n = 58). The PWUC group had to have used cannabis 1–7 days per week, on average, over the past three months. Additionally, young adult PWUC were only eligible if they had *not* used cannabis regularly prior to the age of 18, defined as ≥once per week for a period of ≥3 months, to isolate the impact of adolescent cannabis use in age-group comparisons. Controls had to have used cannabis or tobacco at least once, to match groups on potential confounders associated with the opportunity to use drugs. They could have a maximum of 10 lifetime uses of cannabis and no use in the month prior to the baseline session.

Young adults were aged 26–29 years and adolescents were aged 16–17 years. Exclusion criteria for all participants were: daily use of any psychotropic medication, past-month treatment for a mental health condition (including CUD), and use of any one illicit drug more than twice per month over the past three months. All inclusion and exclusion criteria were assessed at baseline and a comprehensive list can be found in the full study protocol[32]. All participants provided written informed consent to participate. The study was approved by the University College London ethics committee, project ID 5929/003, and conducted in line with the Declaration of Helsinki.

### Materials
CUD level was both a predictor (baseline) and outcome (follow-up) and was assessed with the Mini Neuropsychiatric interview for DSM-5 CUD (CUD-MINI). This is a researcher-administered 11-item measure with questions addressing cannabis-related problems over the past 12 months, such as "have you had cravings or a strong desire or urge to use cannabis?" and "have you continued to use cannabis even though it was clear that the cannabis had caused or worsened psychological or physical problems?"[33,34]. Based on the number of items endorsed, participants were classified into categories of no (0-1 symptoms), mild (2–3 symptoms), moderate (4–5 symptoms), or severe (6+ symptoms) CUD, consistent with the DSM-5 CUD specifiers. Baseline CUD was included as a binary predictor variable (CUD vs. no CUD) and follow-up CUD was included an ordinal outcome variable (no, mild, moderate, severe CUD). The CUD-MINI had good reliability in the current sample (Cronbach's *alpha* = 0.80, based on $n = 237$ baseline participants).

In addition to CUD, baseline predictors were age-group, gender, frequency of cannabis use, problematic tobacco use, problematic alcohol use, negative life events, and COVID-19 lockdown. Cannabis use frequency was measured as average days per week of use in the past three months at baseline with the timeline follow-back interview[35,36]. Problematic alcohol use was assessed with the Alcohol Use Disorder Identification Test (AUDIT), which is scored from 0 to 40[37]. A score of 0-7 indicates low risk, 8-15 indicates increasing risk, 16-19 indicates high risk, and 20 or more indicates possible dependence. The AUDIT had acceptable reliability in the current sample (Cronbach's *alpha* = 0.73, based on $n = 258$ baseline participants). Problematic tobacco use was assessed with the Heaviness of Smoking Index (HSI), which is a valid and reliable measure of smoking severity[38,39]. The HSI is scored from 0 to 6, with scores of 0–1, 2–4, and 5–6 indicating low, medium, and high nicotine dependence, respectively. Negative life events were measured as the number of events, out of 11, the participant self-reported having experienced in the past year at baseline. Examples included breaking off a steady relationship, serious illness or injury, and death of a loved one. Items were selected based on Brugha's List of Threatening Experiences[40], and are identical to those included in a previous study examining predictors of CUD[41]. Age at baseline was verified with photo identification and categorised as young adult (26–29 years) and adolescent

(16–17 years). Gender was self-reported by participants from the options 'male', 'female', 'other', and 'prefer not to say'. Finally, we assessed whether the 12-month follow-up session took place before or after March 23$^{rd}$ 2020, the date of the first nationwide COVID-19 lockdown in the UK.

## Procedures

Full data collection procedures are outlined in the CannTeen study protocol[32]. All baseline sessions and follow-up sessions taking place on or before March 23rd 2020 (i.e., before the COVID-19 lockdown) were completed in-person at University College London. Remaining follow-up sessions (n = 78/33.6%) took place online. All measures were completed at the baseline session, alongside other measures which are listed in full in the CannTeen pre-registration document[32]. The CUD-MINI was additionally completed at the 12-month follow-up session. Participants completed an instant saliva drugs test (Alere DDSV 703 or ALLTEST DSD-867MET/C), a breathalyser (Lion Alcometer), and self-reported abstinence to confirm no recent use of alcohol or cannabis (≥12 hours cut-off) or illicit drugs (≥48 hours cut-off) at the start of all in-person study sessions. Participants with a blood alcohol content reading >0 or a positive result for or self-report of recent use of any illicit drug (including cannabis) were rescheduled. For the virtual study sessions, drug and alcohol abstinence was assessed using self-report only.

## Statistics and reproducibility

Analyses were pre-registered to the Open Science Framework prior to receiving the data[23] and performed using R 4.4.1 and SPSS. The data were first inspected to ensure model assumptions were met. The associations between baseline Age-Group, Gender, Cannabis Use Frequency, CUD, AUDIT, HSI, Negative Life Events, and COVID-19 Lockdown with the outcome follow-up CUD Level were then investigated using cumulative odds ordinal logistic regression with proportional odds. These factors were selected a priori based on their putative association with CUD as outlined in the introduction and tested together in the same model. The model was repeated within participants who were categorised as PWUC at baseline (n = 117) in an exploratory analysis, to investigate which factors predicted CUD within PWUC specifically. All statistical tests were two-sided.

The CannTeen study was powered to detect a difference in CUD between young adult and adolescent PWUC[32]. With 232 participants and a Bonferroni-corrected alpha-level of 0.05, we had 83.7% power to detect a small-to-medium effect size of OR = 2.575[42] for each predictor. This was calculated using the Hmisc R package[43] and is an approximation only, as the function used assumes a balanced two-group structure for the independent variable.

## Reporting summary

Further information on research design is available in the Nature Portfolio Reporting Summary linked to this article.

## Results

### Participant characteristics

Participant characteristics by CUD level at follow-up are presented in Table 1. There were no differences between participants that did (n = 234) and did not (n = 40) complete the follow-up assessment in Gender ($X^2$ = 1.74, p = 0.19), Age-Group ($X^2$ = 0.01, p = 0.92), Cannabis Use Frequency (t = 0.12, p = 0.90), AUDIT score (t = 0.12, p = 0.91), HSI score (t = 0.36, p = 0.72), or past-year Negative Life Events (t = 1.58, p = 0.12). However, there was a difference in Baseline CUD ($X^2$ = 5.14, p = 0.02) with a relatively higher proportion of participants with no CUD (89.9%) completing the follow-up compared with participants with CUD (80.2%). Number of days from baseline to follow-up ranged from 318 to 479, with a mean of 377 (standard deviation = 19).

### Main results

Full results are presented in Tables 2 and 3. The ordinal logistic regression models showed that both Age-Group and baseline CUD were significant predictors of CUD Level at follow-up (Fig. 1). Adolescents had higher likelihood than young adults (OR = 3.26, p < 0.001, 95% confidence interval, CI = 1.65–6.42) and participants with CUD at baseline had higher likelihood than participants with no CUD at baseline (OR = 45.15, p < 0.001, 95% CI = 14.70–138.60) to have higher CUD level at follow-up. None of the other variables in our models were statistically significant predictors of follow-up CUD Level (all p's ≥ 0.08).

Results did not change when the analysis was repeated within the PWUC group only, although the OR for baseline CUD was reduced (adolescents OR = 3.82, p < 0.001; baseline CUD OR = 9.48, p < 0.001). Within PWUC, adolescents and young adults were equally likely to increase and decrease in CUD Level from baseline to follow-up; 25% in both age-groups (n = 15 adolescents, n = 14 young adults) showed an increase and 25% in both age-groups (n = 15 adolescents, n = 14 young adults) showed a decrease (Fig. 2). However, more adolescents (n = 11/18%) than young adults (n = 3/5%) changed from no, mild, or moderate CUD at baseline to severe CUD at follow-up. Given that earlier age of onset might increase risk even within discrete age-groups, we ran additional sensitivity analyses controlling for age of first cannabis use (see Supplementary Table 1). This did not change the pattern of results.

## Discussion

The current longitudinal study explored which factors predicted CUD at one-year follow-up in a sample of young adults and adolescents who did and did not use cannabis regularly at baseline. Baseline CUD and adolescent age were both significantly associated with greater follow-up CUD levels, both in the full sample and in PWUC. Gender, frequency of cannabis use, problematic alcohol use, problematic tobacco use, COVID-19 lockdown, and negative life events were not significant predictors of CUD.

Our results are consistent with previous research[3,4,44–49], which has shown that adolescent PWUC have ~3 times greater odds of having CUD compared with young adult PWUC. The current study adds to these results by showing that over one year, when adjusting for CUD status at baseline, adolescents had more than three times the odds of having higher CUD levels at follow-up compared with young adults. Adolescent age was a significant predictor both in the full sample and in only those who used cannabis at baseline. Within the PWUC group, more adolescents than young adults changed from no, mild, or moderate CUD levels at baseline to severe CUD at follow-up (Fig. 2). In fact, 39% of the adolescents (n = 7) but none of the young adults with no or mild CUD at baseline had severe CUD at follow-up. Therefore, in addition to having increased odds of CUD at follow-up, more adolescents than young adult PWUC transitioned to severe CUD over the year.

As our models included cannabis use frequency, the age-group difference cannot be attributed to the adolescents using more cannabis than the young adults. Indeed, age-group was the only significant predictor, other than baseline CUD, in models which included several other factors hypothesised to confer greater risk of CUD, suggesting that adolescent age is a highly robust predictor of CUD levels over and above other risk factors. There are numerous possible reasons for this observed adolescent vulnerability, including ongoing neurodevelopment[50], differing psychological profiles with greater novelty seeking and lower self-regulation[51], and social factors such as peer influence and having fewer social and lifestyle restrictions on using cannabis[52]. Future studies should explore to what extent these factors predict CUD among PWUC in general, and among adolescent PWUC in particular. It is worth noting that a younger age of onset is also a risk factor for other substance use disorders and is therefore not unique to CUD[45,53]. Importantly, in the absence of other risk factors, adolescent substance use disorders typically resolve naturally by early adulthood without formal treatment or intervention[15,54].

As expected, CUD at baseline predicted CUD level at follow-up. The two most "extreme" categories (no CUD and severe CUD) were relatively stable over time, with roughly 92% of participants with no CUD and 69% of participants with severe CUD at baseline being in the same category at follow-up. Moreover, only four people (3%) with no CUD at baseline had

**Table 1 | Participant characteristics at baseline by CUD level at follow-up**

| | No CUD (n = 133) | Mild CUD (n = 26) | Moderate CUD (n = 28) | Severe CUD (n = 45) |
|---|---|---|---|---|
| **Cannabis group** | | | | |
| PWUC | 22 (16.5%) | 23 (88.5%) | 27 (96.4%) | 45 (100%) |
| Control | 111 (83.5%) | 3 (11.5%) | 1 (3.6%) | 0 |
| **Ethnicity n (%)** | | | | |
| White | 88 (66.2%) | 16 (61.5%) | 15 (53.6%) | 30 (66.7%) |
| Mixed | 13 (9.8%) | 5 (19.2%) | 6 (21.4%) | 5 (11.1%) |
| Asian | 22 (16.5%) | 2 (7.8%) | 3 (10.7%) | 5 (11.1%) |
| Black | 3 (2.3%) | 2 (7.8%) | 3 (10.7%) | 4 (8.9%) |
| Other | 5 (3.8%) | 1 (3.8%) | 1 (3.6%) | 1 (2.2%) |
| Prefer not to say | 2 (1.5%) | 0 | 0 | 0 |
| **Maternal education** | | | | |
| Below undergraduate degree | 57 (43.8%) | 13 (54.2%) | 21 (75.0%) | 23 (51.1%) |
| Undergraduate degree or above | 73 (56.2%) | 11 (45.8%) | 7 (25.0%) | 22 (48.9%) |
| Missing | 3 | 2 | | |
| Days/week alcohol use | 1.22 (1.18) | 0.79 (0.69) | 0.93 (1.21) | 0.79 (0.93) |
| Days/week cigarette/ roll-up use | 0.63 (1.71) | 1.47 (2.57) | 2.01 (2.81) | 1.74 (2.51) |
| **Any other illicit drug used ≥monthly, past 3 months** | | | | |
| Yes | 9 (6.8%) | 5 (19.2%) | 10 (35.7%) | 28 (62.2%) |
| No | 124 (93.2%) | 21 (80.8%) | 18 (64.3%) | 17 (37.8%) |
| **Model predictors** | | | | |
| **Gender**[a] | | | | |
| Female | 70 (52.6%) | 16 (61.5%) | 16 (57.1%) | 18 (40.0%) |
| Male | 63 (47.4%) | 10 (38.4%) | 12 (42.9%) | 27 (60.0%) |
| **Age-Group** | | | | |
| Adolescents (16–17 years) | 58 (43.6%) | 13 (50.0%) | 14 (50.0%) | 33 (73.3%) |
| Young adults (26–29 years) | 75 (56.4%) | 13 (50.0%) | 14 (50.0%) | 12 (26.7%) |
| **Baseline CUD level** | | | | |
| No CUD | 122 (91.7%) | 7 (26.9%) | 3 (10.7%) | 1 (2.2%) |
| Mild CUD | 7 (5.3%) | 8 (30.8%) | 9 (32.1%) | 6 (13.3%) |
| Moderate CUD | 1 (0.8%) | 7 (26.9%) | 9 (32.1%) | 7 (15.6%) |
| Severe CUD | 3 (2.3%) | 4 (15.4%) | 7 (25.0%) | 31 (68.9%) |
| Average days/week of cannabis use, past 3 months | 0.67 (1.71) | 3.38 (2.11) | 4.11 (2.09) | 4.38 (1.90) |
| AUDIT score | 5.21 (3.91) | 5.38 (4.87) | 6.00 (4.27) | 7.16 (4.76) |
| HSI score | 0.05 (0.31) | 0.15 (0.54) | 0.29 (0.76) | 0.20 (0.55) |
| Past-year negative life events | 1.96 (1.51) | 2.38 (1.90) | 2.14 (1.84) | 2.09 (1.70) |
| **Completed follow-up during COVID-19 lockdown** | | | | |
| Yes | 49 (36.8%) | 7 (26.9%) | 14 (50.0%) | 8 (17.8%) |
| No | 84 (63.2%) | 19 (73.1%) | 14 (50.0%) | 37 (82.2%) |

[a]Participants could also select 'other' and 'prefer not to say' but no participants chose these options.
*AUDIT* Alcohol Use Disorder Identification Test, *COVID* coronavirus disease, *CUD* cannabis use disorder, *HSI* Heaviness of Smoking Index, *PWUC* people who use cannabis.
Mean (standard deviation) presented unless otherwise indicated.

either moderate or severe CUD at follow-up and seven people (16%) with severe CUD at baseline had either mild or no CUD at follow-up. Overall, roughly the same proportion of participants increased (*n* = 33/14%) and decreased (*n* = 29/12.5%) in the CUD category from baseline to follow-up, likely due to the myriad factors that may contribute to an escalation or decline in symptoms over a year. It is notable that baseline frequency of

cannabis use was not a predictor of future CUD. Although greater cannabis use has been identified as a risk factor based on other studies[4,55,56], our results suggest that using cannabis more frequently does not necessarily predict future CUD over and above baseline CUD and other predictors. It is also worth keeping in mind that while most people who have CUD use cannabis frequently, many people who use cannabis frequently do not have CUD.

The current results suggest that CUD levels did not differ between males and females. This contrasts with several previous studies, which have found that males are more likely to have CUD[3,10–12,15,56]. However, most of these studies included people who did and did not use cannabis, which means that a greater propensity to use cannabis among males may partly explain this difference. Contrary to this, the current study purposively recruited balanced groups of males and females based on their cannabis use levels. Previous studies comparing gender differences in CUD within PWUC only have shown mixed results, with some finding no differences[41], consistent with the current results, and others still finding a greater risk in males[46,57]. It is also possible that the current study did not have sufficient power (84% power to detect a small-to-medium OR = 2.6) to detect the typically small (OR~2) association between gender and CUD. Crucially, the association between sex, gender, and substance use disorders is not unidimensional but may depend on the symptom or stage (e.g., acquisition *versus* escalation) and type of difference (e.g., qualitative or quantitative) explored[58,59]. Therefore, more research is needed to determine the existence, cause, and nature of sex and gender differences in CUD.

Contrary to our hypotheses, we also did not find a significant association between problematic alcohol or tobacco use at baseline and CUD at follow-up. This is inconsistent with several previous studies, including large longitudinal and nationally representative cohort studies, which have found that both use of and dependence on alcohol and tobacco predict CUD[3,12,13,56,60,61]. It is possible that our sample was too small to replicate these findings, although our study was powered to detect effects in the upper range of what has previously been found for problematic alcohol use (OR~2.5) and tobacco use (OR~4). A more likely explanation is that we had too few participants with high-risk alcohol or tobacco use. Most participants in our sample were not regular smokers and 95% scored <3 on the HSI, indicating low levels of nicotine dependence. Similarly, 72.4% of participants scored <8 on the AUDIT, indicating low risk of alcohol dependence. However, as most of our participants did consume alcohol (93.5% scored >0 on the AUDIT), our results suggest that more problematic alcohol use is not always associated with CUD.

Finally, neither the COVID-19 lockdown nor past-year negative life events were associated with a greater risk of CUD in our sample. Negative or stressful life events have been associated with problematic cannabis in some[16,41,62] but not all[57] previous studies. To our awareness, only one previous longitudinal study has explored whether CUD rates changed during the COVID-19 lockdown[63]. This study found that CUD symptom severity remained relatively stable in a community sample of 120 PWUC, consistent with the current results. Future representative and longitudinal studies are needed to determine whether and what types of adverse experiences in adolescence and adulthood increase the risk of CUD.

A key strength of the current study was the longitudinal design with baseline measures of CUD. This meant that we could test predictors that preceded the CUD outcome, although it is still not possible to say whether any of the factors measured are causative of CUD. Another important strength was the large number of adolescent PWUC, which is typically a more challenging group to recruit compared with adult PWUC, and is therefore relatively underrepresented in cannabis research. Other strengths include the rigorous assessment of CUD using validated DSM-5 diagnostic criteria, ensuring appropriate sensitivity and specificity in diagnosing CUD, balanced age and gender groups, pre-registration of analyses and hypotheses, high levels of participant retention, and data collection both before and during the first COVID-19 lockdown.

The CannTeen study used purposive sampling to recruit balanced groups of PWUC and controls, and within these groups, equal numbers of

**Table 2 | Results of ordinal logistic regressions predicting CUD Level (no, mild, moderate, severe) by baseline factors in the full sample of *n* = 232 participants**

|  | *B* | Standard error | *p* | Odds ratio | 95% CI, lower | 95% CI, upper |
|---|---|---|---|---|---|---|
| Female | −0.08 | 0.33 | 0.80 | 0.92 | 0.48 | 1.75 |
| Male | Reference group | | | | | |
| Adolescent | 1.18 | 0.35 | <0.001 | 3.26 | 1.65 | 6.42 |
| Young adult | Reference group | | | | | |
| CUD | 3.81 | 0.57 | <0.001 | 45.15 | 14.70 | 138.60 |
| No CUD | Reference group | | | | | |
| After COVID-19 lockdown | −0.62 | 0.36 | 0.09 | 0.56 | 0.27 | 1.15 |
| Before COVID-19 lockdown | Reference group | | | | | |
| Days/week cannabis use | 0.18 | 0.10 | 0.08 | 1.20 | 0.98 | 1.47 |
| AUDIT score | 0.04 | 0.04 | 0.30 | 1.04 | 0.97 | 1.12 |
| HSI score | 0.03 | 0.30 | 0.93 | 1.03 | 0.57 | 1.85 |
| Negative life events | 0.02 | 0.10 | 0.83 | 1.02 | 0.85 | 1.23 |

*AUDIT* Alcohol Use Disorder Identification Test, *COVID* coronavirus disease, *CUD* cannabis use disorder, *HSI* Heaviness of Smoking Index.

**Table 3 | Results of ordinal logistic regressions predicting CUD Level (no, mild, moderate, severe) by baseline factors in *n* = 117 participants who used cannabis 1-7 days per week at baseline**

|  | *B* | Standard error | *p* | Odds ratio | 95% CI, lower | 95% CI, upper |
|---|---|---|---|---|---|---|
| Female | −0.21 | 0.36 | 0.56 | 0.81 | 0.40 | 1.64 |
| Male | Reference group | | | | | |
| Adolescent | 1.34 | 0.38 | <0.001 | 3.82 | 1.83 | 7.98 |
| Young adult | Reference group | | | | | |
| CUD | 2.25 | 0.58 | <0.001 | 9.48 | 3.03 | 29.65 |
| No CUD | Reference group | | | | | |
| After COVID-19 lockdown | −0.45 | 0.39 | 0.25 | 0.64 | 0.30 | 1.37 |
| Before COVID-19 lockdown | Reference group | | | | | |
| Days/week cannabis use | 0.02 | 0.11 | 0.85 | 1.02 | 0.83 | 1.25 |
| AUDIT score | 0.01 | 0.04 | 0.73 | 1.01 | 0.94 | 1.10 |
| HSI score | 0.01 | 0.31 | 0.97 | 1.01 | 0.56 | 1.85 |
| Negative life events | 0.03 | 0.10 | 0.76 | 1.03 | 0.84 | 1.26 |

*AUDIT* Alcohol Use Disorder Identification Test, *COVID* coronavirus disease, *CUD* cannabis use disorder, *HSI* Heaviness of Smoking Index.

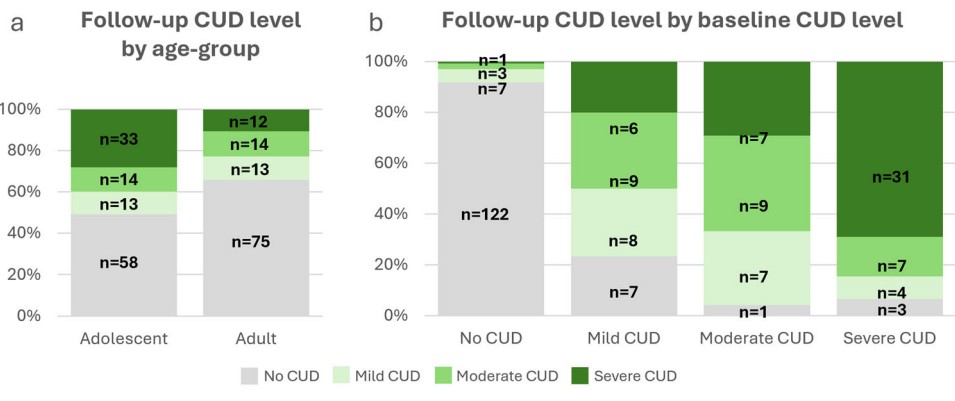

**Fig. 1 | Follow-up cannabis use disorder (CUD) Level by Age-Group and baseline CUD level in *n* = 232 participants.** Follow-up CUD level by **a** age-group and **b** baseline CUD level. An ordinal logistic regression found that adolescent age and baseline CUD significantly predicted higher CUD levels at follow-up.

young adults (26–29 years) and adolescents (16–17 years) and males and females. While this was necessary to ensure a sufficient number of people who used cannabis frequently in both age and gender groups, it inevitably reduced the representativeness of our sample. This is important to consider when interpreting the current results, as risk factors for CUD may vary between groups with different backgrounds and circumstances. Second, there are several potential risk factors for CUD which the current study did not test, including sociodemographic, social (e.g., family situation, peer environment), psychological, and psychiatric factors, and other substance use and substance use disorders[3,8,12,13,56]. Furthermore, the legal status of

**Fig. 2 | Changes in levels of cannabis use disorder (CUD) from baseline to follow-up in** $n = 117$ **participants who used cannabis 1–7 days/week at baseline, by Age-Group.** Follow-up CUD level by baseline CUD level in **a** adolescents and **b** adults.

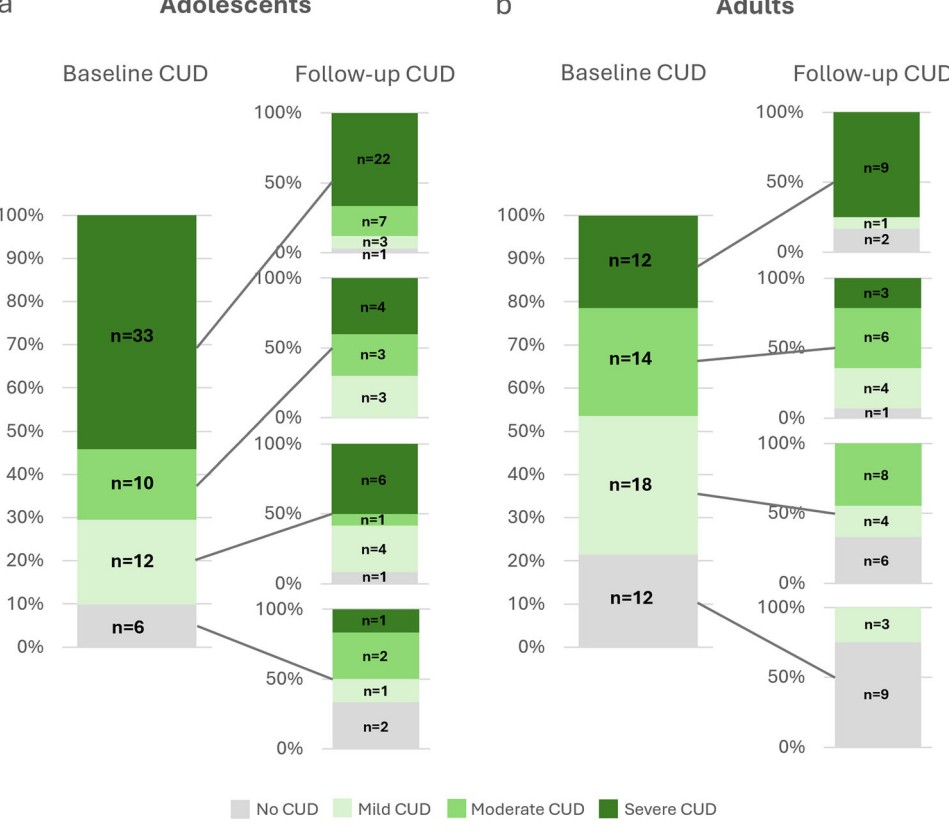

cannabis may affect the prevalence of CUD[64–66] and future studies should explore the impact of different cannabis policies on risk factors for CUD. Third, given that the majority of participants who qualified for CUD at follow-up also did so at baseline, the current study should not be interpreted as exploring the antecedents of CUD. Future large-scale cohort studies in people who have not yet started using cannabis are needed to do this. Follow-up after 12 months is also needed to explore developments that occur over a longer period of time. Finally, relatively more participants with no CUD at baseline completed the follow-up compared with participants with CUD. However, given that this was a small difference and retention was high (>80%) in both groups, we do not believe this introduced serious bias to our results.

Consistent with our hypotheses and previous studies, the current study found that adolescent age significantly predicted higher CUD levels at 12 months, both in the full sample and among only those who used cannabis. Gender, cannabis use frequency, problematic alcohol use, problematic tobacco use, more negative life events, and COVID-19 lockdown did not predict 12-month CUD levels in this study. While some adolescents naturally 'mature out' of substance use disorders, others do not, and adolescent PWUC may still experience harmful effects of CUD even if they later remit. Therefore, treatment and prevention interventions should target adolescent PWUC.

## Data availability

The participants of this study did not give written consent for their data to be shared publicly and the data supporting this research are therefore not available. The source data underlying Figs. 1 and 2 can be found within the figures and are displayed as frequencies.

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

## Acknowledgements

The CannTeen study was funded by a grant from the Medical Research Council (MRC; award number MR/P012728/1) to HVC and TPF. We thank Sharinjeet Dhiman for all her work on the project. We are also thankful to all participants in the CannTeen study and to all those who contributed to data collection.

## Author contributions

V.C. and T.P.F. secured funding for this study. C.M., T.P.F., V.C. and W.L. contributed to the conception and C.M., D.J., M.S., T.P.F., V.C., W.L., and D.J. contributed to the design of the study. C.M., K.P., M.S., R.L., S.O., and W.L. contributed to data collection. D.J., M.S., and T.P.F. were responsible for analyses and interpretation. M.S. and D.J. drafted the original manuscript, and all authors critically reviewed and approved the manuscript as submitted.

## Competing interests

HVC has consulted for Janssen Research and Development. The remaining authors declare no competing interests.
