## [Peer review file · Communications Medicine]

Longitudinal study of risk factors predicting cannabis use disorder in UK young adults and adolescents

Corresponding Author: Dr Martine Skumlien

Version 0:

Reviewer comments:

Reviewer #1

(Remarks to the Author)

Thank you for inviting me to review the manuscript entitled "Which factors predict cannabis use disorder? A 12-month longitudinal study in adults and adolescents who do and do not use cannabis."

I have a few suggestions and questions regarding the manuscript: 1) It would be beneficial to include information on cannabis use policies in the UK to provide context for the study. In the present form, the manuscript does not clearly explain how the eight factors were selected. Did the authors conduct a prediction study assessing model parameters to choose these factors? And the term "pre-registered" is unclear. Did the authors conduct a pre-screening of factors associated with cannabis use disorder (CUD) from the study website? Please clarify this. 2) Study Location: Initially, the study appears to be Canadian, but it is actually based in the UK according to the review committee. Please clarify the location/country in the design and methods section. Recruitment: The age groups included in the study (16-17 and 26-29 years old) are unusual. What was the rationale for excluding individuals aged 18-25? This age group often exhibits similar behaviors to adolescents regarding cannabis use and should have been considered unless justified. 3) Analysis concerns: As presented in table, the cell sizes for levels of CUD are too small to perform ordinal logistic regression, and the reported odds ratios (ORs) of 100+ and wide CIs seem unrealistic to me. I suggest combining the CUD levels into one category versus not having CUD and re-analyzing the data. Additionally, this study is focused on association (etiologic research) rather than prediction, so it should be framed accordingly.

Reviewer #2

(Remarks to the Author)

The present investigation examined eight predictors of Cannabis Use Disorder (CUD) using a longitudinal design and purposive sampling to recruit both adults and adolescents who currently use cannabis including those who do not use cannabis. CUD was assessed using the M.I.N.I. CUD module for DSM-5. For predictors other than baseline CUD, age and gender, the others used validated self-report measures to assess frequency of cannabis use, problematic alcohol and tobacco use, negative life events, and COVID-19 lockdown. Of the eight pre-registered predictors, only adolescent age and CUD (mild, moderate, and severe [relative to no CUD]) predicted CUD at follow-up. Overall, the manuscript is organized, well-written, and addresses an important question. Below, please find my comments addressed to the attention of the authors.

Major Concern(s): N/A

Minor Concern(s):

Title and Abstract

1. The authors should consider whether there's any value in listing the location of the study in the title and the abstract.

Introduction

2. Roughly half of the opening paragraph is spent listing DSM-5 criteria for CUD, which could be shortened. Shortening the

description of CUD would give the authors some room to list the mental health, cognitive, and physical problems that accompany CUD.

3. The authors list psychosocial and pharmacologic interventions for CUD. Do behavioral interventions (e.g., contingency management) fall under the umbrella of psychosocial interventions here?
4. Of the potential risk factors for CUD that have been identified, it may be helpful to list the magnitude of these predictors, such as the odds ratios observed in those studies.
5. It would be useful to briefly list why findings have not been consistent across studies to educate the reader.
6. After listing the negative impact of adverse childhood experiences, the authors state how negative experiences later in life could affect CUD risk. As there is likely some overlap between adverse childhood experiences and negative experiences later in life, it would be helpful to include a few examples of each. In addition, it would be helpful to provide an operational definition for adverse childhood experiences and negative life experiences
7. The authors' pre-registered hypotheses is a notable strength of this study.
8. Out of curiosity, did the authors have a rationale for examining cannabis frequency, but not cannabis quantity? I know this issue continues to be a challenge for the field, but did the authors consider a measure like Cutler et al.'s (DFAQ-CU)?

Methods

9. What's the current legal status of cannabis in the United Kingdom?
10. For the sentence that details the four CUD categories, I recommend clarifying that these categories are consistent with the DSM-5's CUD specifiers.
11. For the TLFB and cannabis use frequency, did participants specify whether they used alone or with other peers in a social setting (as quantity may decrease in the presence of peers who also use).
12. Is the Heaviness of Smoking Index (HSI) suitable for assessing use of combustible tobacco and e-cigarettes?
13. In the assessment of negative life events, I can imagine that there's a decent amount of variability in terms of what's being reported. Were there ever any instances that were ambiguous? If so, how were they resolved? On the other hand, my question might be irrelevant here as judging the negative life events through the eyes of the researcher and through the eyes of the participant are not one and the same.
14. Please specify the names of the measures that were reported elsewhere.
15. Please specify the manufacturers (and their respective locations) for the instant saliva drug test and the breathalyzer.

Results

16. The results are easy to interpret and nicely organized.
17. Please report percentages in addition to sample sizes to help with interpretations (in the main text, where applicable).
18. It would be useful to include a p-value (e.g., all p 's > xxx) at the end of the statement, "None of the other variables in our models were statistically significant predictors of follow-up CUD."
19. For both Figure 1 and Figure 2, adding the sample size into the respective follow-up CUD boxes (and not just baseline) would help with interpretation.
20. What do the authors make of some of the extremely wide 95% confidence intervals listed in Tables 2 and 3? In particular, mild, moderate, and severe CUD.

Discussion

21. The number of participants who changed from no, mild, or moderate CUD levels at baseline to severe CUD at follow-up is an interesting finding. Do the authors have any hypotheses for what's going on here, or is this simply an effect of time?
22. The authors' interpretation of the observed reasons for adolescent vulnerability are interesting. In mentioning different psychological profiles that focus on constructs that change over time (i.e., self-regulatory ability, impulsivity, sensation seeking), are any of these constructs suitable for constructs for future work attempting to uncover predictors of CUD?

23. To what degree might cannabis legal status and/or cannabis availability/accessibility be other candidate variables to examine as predictors of CUD? Have there been studies in Canada or the USA that might speak to this?

24. This paper is one of the few that I've read recently that does an excellent job assessing CUD with an interview that yields appropriate sensitivity and specificity relative to the diagnosis of CUD (and may be worth noting as a strength).

Reviewer #3

(Remarks to the Author)

This manuscript is well-written and addresses an important need for more longitudinal research examining problematic cannabis use. A significant limitation that is not addressed and therefore limits enthusiasm for the findings is a confounding risk factor of age of first use of cannabis. Earlier initiation of use may confer greater risk during brain/cognitive development, and by excluding adult PWUC if they had used regularly prior to age 18, the interpretation of adolescents being at higher risk could simply be related to earlier initiation of use overall. The authors would need to address this issue more fully prior to publication. More minor comments are listed below.

1. Please consider changing your group name throughout the manuscript to "young adults," as the title led me and might lead other readers to assume a wider age range than 26-29.
2. Abstract, line 48: For clarity, consider updating "only those who used cannabis" to something like "only those who used cannabis at least once per week at baseline."
3. Introduction, line 81 and 98: Given that "dependence" is no longer used in clinical diagnoses, consider changing to "problematic use."
4. Introduction, line 115: Some studies use "did and did not use cannabis" to indicate a comparison group of participants who have never tried cannabis at all. Consider adding regularly/routinely or some other descriptor.
5. Discussion, lines 283-285: Considering your main conclusion of the paper, expanding or clarifying this statement would be helpful; for example, is this also true of severe SUD, or just mild, etc.?

Reviewer #4

(Remarks to the Author)

The authors of this study investigated several risk factors for the development of cannabis use disorder in adolescents and young adults using a longitudinal design with visits taking place one year apart. Based on the findings, they conclude that adolescent age and existing CUD are risk factors for the development of more severe CUD symptoms. While I find the manuscript to be clearly written, I do not find the results and interpretations compelling. To understand the antecedents of CUD, it is difficult to make conclusions when a large percentage of participants already have CUD symptoms at baseline, particularly in the adolescent sample. A significant body of literature already suggests that adolescence is a time of increased risk for the development of substance use disorder, so the conclusions from the findings are not particularly novel. In addition, while longitudinal studies provide a powerful approach for investigating risk factors, the 12-month period between appointments may lead to premature conclusions as SUD development may occur on a different time scale depending on age of participants.

Here are more specific comments for each section of the manuscript:

Introduction:

1) The authors discuss several risk factors that have been well established as increasing vulnerability for the development of CUD and others with mixed findings. It is not clear what the research question is at the end of the Introduction. While the hypotheses are presented based on previous findings, the Introduction should more clearly describe why this study is novel.

Methods:

- 1) Could the authors define: "...had not used cannabis regularly prior to the age of 18..." What is defined as regular use?
- 2) Why were controls required to have used tobacco at least once?
- 3) Please present the substance use characteristics of the sample in Table 1 (e.g., frequency of cannabis use over last 3 months, lifetime cannabis use, lifetime tobacco use for PWUC and controls).
- 4) Please report the internal consistency of the measures used in the current sample.

Results/Discussion:

- 1) Adolescents compared to adults had a greater likelihood to have CUD at follow-up. However, a greater proportion of adolescents relative to adults already had CUD at baseline, so this may be difficult to completely parse out.
- 2) This study does not allow us to understand the antecedents of baseline CUD risk.
- 3) The authors themselves acknowledge that the small sample size of this study may have led to different findings relative to previous research (e.g., how problematic alcohol or tobacco use may be predictive of CUD). The issues with small sample size in general may lead to inaccurate conclusions regarding the role of sex and frequency of cannabis use at baseline as predictors of CUD at follow-up.

Version 1:

Reviewer comments:

Reviewer #1

(Remarks to the Author)

Dear authors,

Thank you for addressing the comments. I think the manuscript has been improved, but I suggest moving the cannabis policies from the methods section to somewhere in the introduction. Additionally, if any, please include medical cannabis use policies as well. With these revisions, I recommend the manuscript to be accepted for publication.

Reviewer #2

(Remarks to the Author)

The authors' revised submission thoroughly addressed all of my comments, and the authors were very thoughtful in their responses to my original review. It is clear that the authors put a lot of effort into this revision evidenced by their responses to all reviewers and their close attention to detail in preparing this resubmission. Accordingly, I have no outstanding comments or concerns and hope to see this in press.

Reviewer #3

(Remarks to the Author)

I find that the authors have appropriately and carefully addressed mine and other reviewers' concerns, and that the manuscript is strengthened compared to the first review.

Reviewer #4

(Remarks to the Author)

The authors provided very thorough responses and revisions to all reviewer comments. I believe the manuscript has been significantly improved as a result of these changes and makes an important contribution to the literature.

Dear Reviewers,

Thank you for your helpful and constructive review of our manuscript. We are grateful for the invitation to respond to your concerns and have revised the manuscript in accordance with your suggestions.

Of note, the revised manuscript clarifies how the eight factors in our models were selected and includes odds ratios for CUD risk factors and a 'cannabis research context statement' specifying study location and legal context. We have additionally changed our baseline CUD predictor to a binary variable (CUD vs. no CUD) which has improved model fit without altering the overall pattern of results or conclusions. Finally, the revised manuscript addresses the large proportion of baseline CUD as a limitation along with other important limitations.

All details of the changes to the manuscript are referenced in our response below and are included in the tracked version of the article file.

Yours sincerely,

Martine Skumlien, on behalf of all authors

Reviewer #1 (Remarks to the Author):

Thank you for inviting me to review the manuscript entitled “Which factors predict cannabis use disorder? A 12-month longitudinal study in adults and adolescents who do and do not use cannabis.”

We would like to thank the Reviewer for their careful review of our manuscript.

I have a few suggestions and questions regarding the manuscript:

1) It would be beneficial to include information on cannabis use policies in the UK to provide context for the study. In the present form, the manuscript does not clearly explain how the eight factors were selected. Did the authors conduct a prediction study assessing model parameters to choose these factors? And the term “pre-registered” is unclear. Did the authors conduct a pre-screening of factors associated with cannabis use disorder (CUD) from the study website? Please clarify this.

To address the Reviewer’s first point, we have added a ‘cannabis research context’ section, as recommended in a recent editorial by Cousijn et al. (<https://doi.org/10.1111/add.16460>).

Page 4, line 128-136: *“This study was conducted in London, United Kingdom, between 2017 and 2020. Recreational cannabis is illegal under the Misuse of Drugs Act 1971. The most common method of consumption in the UK was smoking cannabis in a joint together with tobacco and the most prevalent type of cannabis was sinsemilla or ‘skunk’, which contains 14-15% Δ^9 -tetrahydrocannabinol and negligible levels of cannabidiol [25-29]. The past-year prevalence of cannabis use in adults was estimated at 7.8% in 2019/2020, the highest since 2007 (8.2%) [30]. In 2019/2020, 19.6% of first-time presentations to substance use treatment among adults was for cannabis [31].”*

To address the second point, we have added a sentence specifying model selection to the ‘Analyses’ section. We did not conduct any prediction study to select model parameters.

Page 6, line 222-226: *“The associations between baseline Age-Group, Gender, Cannabis Use Frequency, CUD, AUDIT, HSI, Negative Life Events, and COVID-19 Lockdown with the outcome follow-up CUD Level were then investigated using cumulative odds ordinal logistic regression with proportional odds. These factors were selected a priori based on their putative association with CUD as outlined in the introduction and tested together in the same model.”*

‘Pre-registered’ simply refers to our publishing the hypotheses and analyses online on the Open Science Framework prior to looking at the data (as explained under ‘Analyses’) and does not refer to pre-screening of factors.

2) Study Location: Initially, the study appears to be Canadian, but it is actually based in the UK according to the review committee. Please clarify the location/country in the design and methods section. Recruitment: The age groups included in the study (16-17 and 26-29 years old) are unusual. What was the rationale for excluding individuals aged 18-25? This age group often exhibits similar behaviors to adolescents regarding cannabis use and should have been considered unless justified.

We have specified that the study took place in the UK in the new ‘cannabis research context’ section (above).

As noted in the ‘Participants’ section, the CannTeen study was designed to compare the effects of cannabis in adults and adolescents. The age ranges used in this study were chosen for both theoretical and pragmatic reasons. The 16-17-year range for adolescents was chosen because: (i) cannabis use initiation and frequency among adolescents who use cannabis peak at this age; (ii) this is an important time of neurodevelopment; and (iii) 16 years is the minimum age of consent for research participation in the UK. The 26-29-year age range for adults was chosen because: (i) the brain is typically fully matured from age 25; and (ii) the prevalence of cannabis use remains high in this age-group but starts to decline in the early 30s.

3) Analysis concerns: As presented in table, the cell sizes for levels of CUD are too small to perform ordinal logistic regression, and the reported odds ratios (ORs) of 100+ and wide CIs seem unrealistic to me. I suggest combining the CUD levels into one category versus not having CUD and re-analyzing the data. Additionally, this study is focused on association (etiological research) rather than prediction, so it should be framed accordingly.

We are grateful to the Reviewer for drawing our attention to the low cell sizes and in response to their comment we have now re-analysed the data using a binary variable for baseline CUD (CUD vs. no CUD) instead of an ordinal CUD level variable as the predictor. This has markedly reduced the width of the ORs and CIs. Methods, results, discussion sections have been updated accordingly (please see the full manuscript with tracked changes). The pattern of results remains the same and we have therefore not made changes to our interpretations or conclusions.

Table 2 (excerpt):

	B	Standard error	p	Odds ratio	95% CI, lower	95% CI, upper
CUD	3.81	0.57	<.001	45.15	14.70	138.60
No CUD	Reference group					

Table 3 (excerpt):

	B	Standard error	p	Odds ratio	95% CI, lower	95% CI, upper
CUD	2.25	0.58	<.001	9.48	3.03	29.65
No CUD	Reference group					

In reference to the Reviewer’s second point: as our study aims to explore the association between baseline factors and CUD 12 months later, rather than the causal mechanisms of CUD, we believe it meets the criteria for prediction research rather than etiological research as defined in the literature (e.g., van Diepen 2017, <https://doi.org/10.1093/ndt/gfw459>). We have added a sentence to the ‘Strengths and limitations’ section clarifying that we did not aim to explore the antecedents of CUD and that we have been careful to describe all relationships as associative rather than causative.

Page 13, line 403-405: *“Third, given that the majority of participants who qualified for CUD at follow-up also did so at baseline, the current study should not be interpreted as exploring the antecedents of CUD.”*

Reviewer #2 (Remarks to the Author):

The present investigation examined eight predictors of Cannabis Use Disorder (CUD) using a longitudinal design and purposive sampling to recruit both adults and adolescents who currently use cannabis including those who do not use cannabis. CUD was assessed using the M.I.N.I. CUD module for DSM-5. For predictors other than baseline CUD, age and gender, the others used validated self-report measures to assess frequency of cannabis use, problematic alcohol and tobacco use, negative life events, and COVID-19 lockdown. Of the eight pre-registered predictors, only adolescent age and CUD (mild, moderate, and severe [relative to no CUD]) predicted CUD at follow-up. Overall, the manuscript is organized, well-written, and addresses an important question. Below, please find my comments addressed to the attention of the authors.

Thank you very much for your thorough review and positive comments.

Major Concern(s): N/A

Minor Concern(s):

Title and Abstract

1. The authors should consider whether there's any value in listing the location of the study in the title and the abstract.

We have now added this to the title and abstract.

Title: *"Which factors predict cannabis use disorder? A 12-month longitudinal study in young adults and adolescents who do and do not use cannabis in the UK"*

Abstract: *"Participants were 232 young adults (26-29 years) and adolescents (16-17 years) who took part in both the baseline and 12-month follow-up sessions of the London-based 'CannTeen' study."*

Introduction

2. Roughly half of the opening paragraph is spent listing DSM-5 criteria for CUD, which could be shortened. Shortening the description of CUD would give the authors some room to list the mental health, cognitive, and physical problems that accompany CUD.

We have amended this paragraph according to the Reviewer's suggestions.

Page 3, line 79-84: *"The fifth edition of the Diagnostic and Statistical Manual of Mental Disorders (DSM-5) defines CUD as problematic cannabis use leading to clinically significant distress and/or impairment in functioning. Symptoms include using more cannabis than intended, an inability to stop use despite harm, and tolerance and withdrawal symptoms. Individuals with CUD may also experience reduced quality of life alongside higher risk of mental health, cognitive, and physical problems such as mood and psychotic disorders [2]."*

3. The authors list psychosocial and pharmacologic interventions for CUD. Do behavioral interventions (e.g., contingency management) fall under the umbrella of psychosocial interventions here?

We have changed this sentence to “behavioural and psychosocial”.

4. Of the potential risk factors for CUD that have been identified, it may be helpful to list the magnitude of these predictors, such as the odds ratios observed in those studies.

We have amended this paragraph according to the Reviewer’s suggestions.

Page 3, line 96-99: *“In addition to using cannabis frequently and at large quantities, male gender (odds ratio, OR ~2), adolescent age (OR ~3), and more problematic use of other substances such as alcohol (OR ~1.5-2.5) and tobacco (OR ~4, based on one study) have all been identified as risk factors for CUD [2, 10-13].”*

5. It would be useful to briefly list why findings have not been consistent across studies to educate the reader.

We have amended this paragraph according to the Reviewer’s suggestions.

Page 3, line 99-102: *“However, findings have not been consistent across studies, possibly due to differences in design, samples, and measurement tools, and up-to-date replication in different populations and settings is necessary.”*

6. After listing the negative impact of adverse childhood experiences, the authors state how negative experiences later in life could affect CUD risk. As there is likely some overlap between adverse childhood experiences and negative experiences later in life, it would be helpful to include a few examples of each. In addition, it would be helpful to provide an operational definition for adverse childhood experiences and negative life experiences

We have now included examples of adverse childhood experiences and later-life negative experiences in this paragraph.

Page 3, line 103-106: *“For instance, while adverse childhood experiences, such as parental separation, are a relatively consistent predictor of problematic cannabis use and CUD [14-16], it is less clear if negative experiences later in life, such as losing one’s job, increase CUD risk.”*

Additionally, we have included more detail on the negative life events measure in the ‘Materials’ section.

Page 5, line 193-197: “*Negative life events were measured as the number of events, out of 11, the participant self-reported having experienced in the past year at baseline. Examples included breaking off a steady relationship, serious illness or injury, and death of a loved one. Items were selected based on Brugha’s List of Threatening Experiences [40], and are identical to those included in a previous study examining predictors of CUD [41]*”

We have not given a more in-depth definition of adverse childhood experiences, as this was not a predictor in our study.

7. The authors’ pre-registered hypotheses is a notable strength of this study.

We appreciate the Reviewer’s recognition of this.

8. Out of curiosity, did the authors have a rationale for examining cannabis frequency, but not cannabis quantity? I know this issue continues to be a challenge for the field, but did the authors consider a measure like Cutler et al.’s (DFAQ-CU)?

We agree with the Reviewer that measuring quantity is useful to get a more complete picture of cannabis consumption. However, we chose frequency as there is a greater volume of evidence suggesting an association between frequency and CUD risk, whereas the evidence for quantity is more limited at present.

Methods

9. What’s the current legal status of cannabis in the United Kingdom?

We have added a ‘cannabis research context’ section, as recommended in a recent editorial by Cousijn et al. (<https://doi.org/10.1111/add.16460>), which outlines the legal status of cannabis in the UK along with other relevant statistics.

Page 4, line 128-136: “*This study was conducted in London, United Kingdom, between 2017 and 2020. Recreational cannabis is illegal under the Misuse of Drugs Act 1971. The most common method of consumption in the UK was smoking cannabis in a joint together with tobacco and the most prevalent type of cannabis was sinsemilla or ‘skunk’, which contains 14-15% Δ^9 -tetrahydrocannabinol and negligible levels of cannabidiol [25-29]. The past-year prevalence of cannabis use in adults was estimated at 7.8% in 2019/2020, the highest since 2007 (8.2%) [30]. In 2019/2020, 19.6% of first-time presentations to substance use treatment among adults was for cannabis [31].*”

10. For the sentence that details the four CUD categories, I recommend clarifying that these categories are consistent with the DSM-5’s CUD specifiers.

We have amended this sentence according to the Reviewer's suggestions.

Page 5, line 175-178: *"Based on the number of items endorsed, participants were classified into categories of no (0-1 symptoms), mild (2-3 symptoms), moderate (4-5 symptoms), or severe (6+ symptoms) CUD, consistent with the DSM-5 CUD specifiers."*

11. For the TLFB and cannabis use frequency, did participants specify whether they used alone or with other peers in a social setting (as quantity may decrease in the presence of peers who also use).

Yes, participants specified in the TLFB whether they used cannabis alone or with others. However, since frequency and not quantity was used in the models, we have not included this information in the paper.

12. Is the Heaviness of Smoking Index (HSI) suitable for assessing use of combustible tobacco and e-cigarettes?

We have added a reference to the validity and reliability of the HSI. This measure reflects combustible tobacco use only, not e-cigarette use. Only four participants in our sample were daily e-cigarette users.

Page 5, line 190-191: *"Problematic tobacco use was assessed with the Heaviness of Smoking Index (HSI), which is a valid and reliable measure of smoking severity [38, 39]."*

13. In the assessment of negative life events, I can imagine that there's a decent amount of variability in terms of what's being reported. Were there ever any instances that were ambiguous? If so, how were they resolved? On the other hand, my question might be irrelevant here as judging the negative life events through the eyes of the researcher and through the eyes of the participant are not one and the same.

We have included more information on the negative life events measure in the methods (see our response to comment 6). It was up to participants' themselves to determine whether they had experienced each event or not, so no interpretation was required from the researcher.

14. Please specify the names of the measures that were reported elsewhere.

The baseline session of the CannTeen study was 4-5 hours long and included a large number of measures, which are listed in full on the Open Science Framework (osf.io/jg9qp). We have now added text to the manuscript stating that the full list of

measures used in the study are reported here. Given the word count limit of the journal and the potential redundancy of listing many variables that were not included in this analysis, we believe that this is preferable to listing them in the main text.

Page 6, line 208-210: *“All measures were completed at the baseline session, alongside other measures which are listed in full in the CannTeen pre-registration document [32].”*

15. Please specify the manufacturers (and their respective locations) for the instant saliva drug test and the breathalyzer.

We have now added this information.

Page 6, line 211-212: *“Participants completed an instant saliva drugs test (Alere DDSV 703 or ALLTEST DSD-867MET/C), a breathalyser (Lion Alcometer) (...)”*

Results

16. The results are easy to interpret and nicely organized.

Thank you for this positive comment.

17. Please report percentages in addition to sample sizes to help with interpretations (in the main text, where applicable).

We have made sure both percentages and sample sizes are reported.

18. It would be useful to include a p-value (e.g., all p 's > xxx) at the end of the statement, “None of the other variables in our models were statistically significant predictors of follow-up CUD.”

We have now added this information.

Page 8, line 255-256: *“None of the other variables in our models were statistically significant predictors of follow-up CUD (all p 's > .08).”*

19. For both Figure 1 and Figure 2, adding the sample size into the respective follow-up CUD boxes (and not just baseline) would help with interpretation.

We have added follow-up sample sizes to both figures (please see the figure/main article files).

20. What do the authors make of some of the extremely wide 95% confidence intervals listed in Tables 2 and 3? In particular, mild, moderate, and severe CUD.

On suggestion from Reviewer 1, we have changed our baseline CUD predictor to a binary variable (CUD vs. no CUD) to increase the cell sizes for this variable and better meet the requirements of an ordinal logistic regression. This has markedly reduced the ORs and narrowed the CIs without affecting the overall pattern of results.

Table 2 (excerpt):

	B	Standard error	p	Odds ratio	95% CI, lower	95% CI, upper
CUD	3.81	0.57	<.001	45.15	14.70	138.60
No CUD	Reference group					

Table 3 (excerpt):

	B	Standard error	p	Odds ratio	95% CI, lower	95% CI, upper
CUD	2.25	0.58	<.001	9.48	3.03	29.65
No CUD	Reference group					

Discussion

21. The number of participants who changed from no, mild, or moderate CUD levels at baseline to severe CUD at follow-up is an interesting finding. Do the authors have any hypotheses for what's going on here, or is this simply an effect of time?

Participants were equally likely to increase and decrease in CUD category from baseline to follow-up, which we suspect is due to the numerous factors that might contribute to individual changes in CUD over one year. We have now added a sentence on this to the discussion.

Page 11, line 335-338: *“Overall, roughly the same proportion of participants increased (n=33/14%) and decreased (n=29/12.5%) in CUD category from baseline to follow-up, likely due to the myriad factors that may contribute to an escalation or decline in symptoms over a year.”*

22. The authors' interpretation of the observed reasons for adolescent vulnerability are interesting. In mentioning different psychological profiles that focus on constructs that change over time (i.e., self-regulatory ability, impulsivity, sensation seeking), are any of these constructs suitable for constructs for future work attempting to uncover predictors of CUD?

We have now added a sentence on this in this paragraph.

Page 11, line 324-325: *“Future studies should explore to what extent these factors predict CUD among PWUC in general, and among adolescent PWUC in particular.”*

23. To what degree might cannabis legal status and/or cannabis availability/accessibility be other candidate variables to examine as predictors of CUD? Have there been studies in Canada or the USA that might speak to this?

We have now added a sentence on this under ‘Strengths and limitations’.

Page 13, line 401-403: *“Furthermore, the legal status of cannabis may affect the prevalence of CUD [64-66] and future studies should explore the impact of different cannabis policies on risk factors for CUD.”*

24. This paper is one of the few that I’ve read recently that does an excellent job assessing CUD with an interview that yields appropriate sensitivity and specificity relative to the diagnosis of CUD (and may be worth noting as a strength).

Thank you for highlighting this as a strength of our study. We have now added this to the ‘Strengths and limitations’ section.

Page 12, line 387-389: *“Other strengths include the rigorous assessment of CUD using validated DSM-5 diagnostic criteria, ensuring appropriate sensitivity and specificity in diagnosing CUD (...).”*

Reviewer #3 (Remarks to the Author):

This manuscript is well-written and addresses an important need for more longitudinal research examining problematic cannabis use. A significant limitation that is not addressed and therefore limits enthusiasm for the findings is a confounding risk factor of age of first use of cannabis. Earlier initiation of use may confer greater risk during brain/cognitive development, and by excluding adult PWUC if they had used regularly prior to age 18, the interpretation of adolescents being at higher risk could simply be related to earlier initiation of use overall. The authors would need to address this issue more fully prior to publication. More minor comments are listed below.

We thank the Reviewer for their careful evaluation of our manuscript and for their comments.

We agree with the Reviewer that early initiation of cannabis use during important developmental periods may increase the risk of harm. Indeed, this was the key rationale for the overall design of the CannTeen study of exploring whether cannabis use is differentially associated with changes to brain, cognition, and mental health in adolescence and adulthood. Adults were excluded if they had used cannabis regularly before the age of 18 because if both groups used cannabis during adolescence, they

would have the same potential effects from adolescent cannabis use, and our age-group comparisons would not isolate the effect of adolescent cannabis exposure. We refer to this in the 'Participants' section:

Page 4, line 154-156: *“Additionally, adult PWUC were only eligible if they had not used cannabis regularly prior to the age of 18, defined as \geq once per week for a period of \geq 3 months, to isolate the impact of adolescent cannabis use in age-group comparisons.”*

We have run an additional sensitivity analysis controlling for age of first use within the PWUC group, which did not change the pattern of results. We have updated the results section accordingly and included the full results table in the supplemental materials.

Page 8, line 264-266: *“Given that earlier age of onset might increase risk even within discrete age-groups, we ran additional sensitivity analyses controlling for age of first cannabis use (see Supplemental Table 1). This did not change the pattern of results.”*

Our responses to the remaining comments are outlined below.

1. Please consider changing your group name throughout the manuscript to “young adults,” as the title led me and might lead other readers to assume a wider age range than 26-29.

Per the Reviewer's suggestion, we have changed the term to 'young adults' throughout the manuscript.

2. Abstract, line 48: For clarity, consider updating “only those who used cannabis” to something like “only those who used cannabis at least once per week at baseline.”

We have amended this sentence per the Reviewer's suggestion.

Page 2, line 47-49: *“(…) and the COVID-19 lockdown predicted 12-month CUD levels in the full sample and in only those who used cannabis minimum once per week at baseline.”*

3. Introduction, line 81 and 98: Given that “dependence” is no longer used in clinical diagnoses, consider changing to “problematic use.”

We have made this correction in the introduction and throughout the manuscript.

4. Introduction, line 115: Some studies use “did and did not use cannabis” to indicate a comparison group of participants who have never tried cannabis at all. Consider adding regularly/routinely or some other descriptor.

We have amended this sentence per the Reviewer's suggestion.

Page 3, line 118-119: “(...) *in both adolescents and young adults who did and did not use cannabis regularly (...)*”

5. Discussion, lines 283-285: Considering your main conclusion of the paper, expanding or clarifying this statement would be helpful; for example, is this also true of severe SUD, or just mild, etc.?

We have amended this sentence per the Reviewer’s suggestion.

Page 11, line 327-329: “*Importantly, in the absence of other risk factors, adolescent substance use disorders typically resolve naturally by early adulthood without formal treatment or intervention [15, 54].*”

Reviewer #4 (Remarks to the Author):

The authors of this study investigated several risk factors for the development of cannabis use disorder in adolescents and young adults using a longitudinal design with visits taking place one year apart. Based on the findings, they conclude that adolescent age and existing CUD are risk factors for the development of more severe CUD symptoms. While I find the manuscript to be clearly written, I do not find the results and interpretations compelling. To understand the antecedents of CUD, it is difficult to make conclusions when a large percentage of participants already have CUD symptoms at baseline, particularly in the adolescent sample. A significant body of literature already suggests that adolescence is a time of increased risk for the development of substance use disorder, so the conclusions from the findings are not particularly novel. In addition, while longitudinal studies provide a powerful approach for investigating risk factors, the 12-month period between appointments may lead to premature conclusions as SUD development may occur on a different time scale depending on age of participants.

We thank the Reviewer for their thorough review of our manuscript.

As the Reviewer states, longitudinal studies provide a powerful approach for investigating risk factors, and we agree that this is a strength of the current study. To our knowledge, ours is the first study to investigate changes in CUD in matched groups of adolescents and young adults who use cannabis. The 12-month follow up period is well suited to measure changes in CUD symptoms, given the past 12-month time frame of the DSM-5 diagnosis (and as noted by a previous Reviewer, this interview-based diagnosis is another strength of the study). Although a longer follow up period would have been advantageous (we have now noted this as a limitation), overall, we believe that this study offers several key strengths and important contributions to our understanding of CUD.

Page 13, line 406-407: *“Follow-up after 12 months is also needed to explore developments that occur over a longer period of time.”*

We have specified in the limitations that given the high number of people with CUD symptoms at baseline, the current study is suited to explore predictors but not antecedents of CUD. Nonetheless, we believe that it is important to understand the predictors of future CUD levels among people with current CUD, as we have done here.

Page 13, line 403-405: *“Third, given that the majority of participants who qualified for CUD at follow-up also did so at baseline, the current study should not be interpreted as exploring the antecedents of CUD.”*

We acknowledge the Reviewer’s point that the increased risk of CUD among adolescents has been demonstrated previously, and we discuss in the paper. However, the exploration of the effect of COVID-19 lockdown and negative life events in adolescents and young adults is novel and timely. We believe these findings are of interest regardless of whether they are statistically significant and have therefore added them to the conclusion.

Page 13, line 415-417: *“Gender, cannabis use frequency, problematic alcohol use, problematic tobacco use, more negative life events, and COVID-19 lockdown did not predict 12-month CUD levels in this study.”*

Here are more specific comments for each section of the manuscript:

Introduction:

1) The authors discuss several risk factors that have been well established as increasing vulnerability for the development of CUD and others with mixed findings. It is not clear what the research question is at the end of the Introduction. While the hypotheses are presented based on previous findings, the Introduction should more clearly describe why this study is novel.

Per the Reviewer’s suggestion, we have expanded the final section of the introduction to highlight the research questions and the novel aspects of the study.

Page 3-4, line 115-121: *“The current longitudinal study investigated whether eight potential risk factors predicted of CUD levels after one year. We sought to replicate previous findings of relatively well-established risk factors, including age, gender, cannabis use frequency, and problematic alcohol and tobacco use, in both adolescents and young adults who did and did not use cannabis regularly and using a rigorous definition of CUD according to current DSM-5 diagnostic criteria. Additionally, we aimed to test the potential relationship between CUD and less explored risk factors, including negative life events and the COVID-19 lockdown.”*

Methods:

1) Could the authors define: "...had not used cannabis regularly prior to the age of 18..." What is defined as regular use?

We have amended the sentence to define regular use.

Page 4, line 154-155: *"Additionally, adult PWUC were only eligible if they had not used cannabis regularly prior to the age of 18, defined as \geq once per week for a period of \geq 3 months (...)"*

2) Why were controls required to have used tobacco at least once?

We recruited controls with limited cannabis or tobacco exposure rather than no exposure with the aim of more closely matching cannabis users and controls on the opportunity to use drugs and associated unmeasurable confounders. We have added this information to the 'Participants' section.

Page 4, line 156-158: *"Controls had to have used cannabis or tobacco at least once, to match groups on potential confounders associated with the opportunity to use drugs."*

3) Please present the substance use characteristics of the sample in Table 1 (e.g., frequency of cannabis use over last 3 months, lifetime cannabis use, lifetime tobacco use for PWUC and controls).

We do not have good measures of lifetime use of these substances but have included in Table 1 days per week of alcohol and tobacco use. Cannabis use frequency is already included in the table.

Table 1 (excerpt):

	No CUD (n=133)	Mild CUD (n=26)	Moderate CUD (n=28)	Severe CUD (n=45)
Days/week alcohol use	1.22 (1.18)	0.79 (0.69)	0.93 (1.21)	0.79 (0.93)
Days/week cigarette/roll-up use	0.63 (1.71)	1.47 (2.57)	2.01 (2.81)	1.74 (2.51)

4) Please report the internal consistency of the measures used in the current sample.

We have now included this in the 'Materials' section.

Page 5, line 180-181: *"The CUD-MINI had good reliability in the current sample (Cronbach's $\alpha=0.80$, based on $n=237$ baseline participants)."*

Page 5, line 188-190: *“The AUDIT had acceptable reliability in the current sample (Cronbach’s alpha=0.73, based on n=258 baseline participants).”*

Results/Discussion:

1) Adolescents compared to adults had a greater likelihood to have CUD at follow-up. However, a greater proportion of adolescents relative to adults already had CUD at baseline, so this may be difficult to completely parse out.

Given that we controlled for baseline CUD in our analyses, the age-group effect shows the relationship between age-group and CUD over and above baseline CUD levels.

2) This study does not allow us to understand the antecedents of baseline CUD risk.

We now highlight this in the ‘Strengths and limitations’ section.

Page 13, line 403-405: *“Third, given that the majority of participants who qualified for CUD at follow-up also did so at baseline, the current study should not be interpreted as exploring the antecedents of CUD.”*

3) The authors themselves acknowledge that the small sample size of this study may have led to different findings relative to previous research (e.g., how problematic alcohol or tobacco use may be predictive of CUD). The issues with small sample size in general may lead to inaccurate conclusions regarding the role of sex and frequency of cannabis use at baseline as predictors of CUD at follow-up.

We have now amended our discussion to consider these null findings in light of the study’s statistical power and previously observed effect sizes, which we believe addresses the Reviewer’s comment. Typical effect sizes from previous studies are outlined in the introduction.

Page 3, line 96-99: *“In addition to using cannabis frequently and at large quantities, male gender (odds ratio, OR ~2), adolescent age (OR ~3), and more problematic use of other substances such as alcohol (OR ~1.5-2.5) and tobacco (OR ~4, based on one study) have all been identified as risk factors for CUD [2, 10-13].”*

Page 12, line 352-354: *“It is also possible that the current study did not have sufficient power (84% power to detect a small-to-medium OR=2.6) to detect the typically small (OR ~2) association between gender and CUD.”*

Page 12, line 363-365: *“It is possible that our sample was too small to replicate these findings, although our study was powered to detect effects in the upper range of what has previously been found for problematic alcohol use (OR ~2.5) and tobacco use (OR ~4).”*

Dear Reviewers,

Thank you once again for your constructive review of our manuscript. Per Reviewer 1's suggestion, we have moved the 'Cannabis Research Context' section to the introduction and added information about medical cannabis.

We are sincerely grateful to all the Reviewers for their support in improving the manuscript.

Yours sincerely,

Martine Skumlien, on behalf of all authors